# Molecular Surveillance of Canine Degenerative Myelopathy in Breeding Kennels from Romania

**DOI:** 10.3390/ani13081403

**Published:** 2023-04-19

**Authors:** Vlad Cocostîrc, Anamaria Ioana Paștiu, Anca-Alexandra Doboși, Felix Daniel Lucaci, Maria-Carmen Turcu, Mihai Marian Borzan, Dana Liana Pusta

**Affiliations:** 1Department of Genetics and Hereditary Diseases, Faculty of Veterinary Medicine, University of Agricultural Sciences and Veterinary Medicine Cluj-Napoca, 400372 Cluj-Napoca, Romania; 2Department of Animal Breeding and Animal Productions, Faculty of Veterinary Medicine, University of Agricultural Sciences and Veterinary Medicine Cluj-Napoca, 400372 Cluj-Napoca, Romania

**Keywords:** canine degenerative myelopathy (CDM), PCR-RFLP, *SOD-1*, Romanian Bucovina Shepherd, Romanian Mioritic Shepherd, Caucasian Shepherds

## Abstract

**Simple Summary:**

Canine degenerative myelopathy (CDM) is a disease that affects the spinal cord and develops spontaneously in adult dogs. It is caused by a genetic mutation and is inherited, to some extent, by the puppies whose parents are either carriers or affected by the mutation. This survey aimed to determine the presence of the CDM-associated mutation in dogs from different breeding kennels in Romania. The current study included a first screening of the Romanian Bucovina Shepherd, Romanian Mioritic Shepherd, and Caucasian Shepherd breeds. Oral swab samples (*n* = 230) collected from 26 different dog breeds were tested by PCR-RFLP. The mutation was identified in 26 dogs, of which 16 were carriers, being heterozygous for the risk allele, and 10 were homozygous for the risk allele. The mutation was identified in the Wire Fox Terrier, Romanian Mioritic Shepherd, German Shepherd, Rottweiler, Belgian Shepherd, and Czechoslovakian Wolfdog breeds. Genetic testing of dogs for the mutation is an important tool used for the confirmation of affected individuals and preventing the mutation from spreading within the canine population.

**Abstract:**

Canine degenerative myelopathy (CDM) is a spontaneous neurodegenerative disease. Genetically, CDM is an autosomal recessive disease with incomplete penetrance, most commonly caused by a genetic mutation in exon 2 of gene *SOD1* (c.118G > A). This study aimed to determine the mutant allele frequency associated with CDM in various dog breeds from Romania. Dogs (*n* = 230) from 26 breeds were included in the study. Genotyping using the PCR-RFLP technique was performed on DNA extracted from oral swabs. The results revealed that 204 dogs were homozygous for the wild-type allele (G/G), 16 were heterozygous (A/G), and 10 were homozygous for the mutant allele (A/A). The mutant allele was identified in Wire Fox Terrier, Romanian Mioritic Shepherd, German Shepherd, Rottweiler, Belgian Shepherd, and Czechoslovakian Wolfdog breeds. The mutant allele frequency (A) within the tested population was 0.0783. The results for Belgian Shepherd, German Shepherd, and Romanian Mioritic Shepherd were in Hardy–Weinberg equilibrium, but a departure was observed for Rottweiler. The current study included a first screening of the Romanian Bucovina Shepherd, Romanian Mioritic Shepherd, and Caucasian Shepherd breeds. Genetic testing for the mutation associated with CDM is important in order to avoid the risk of the emergence of dogs homozygous for the *SOD1*:c118G > A allele.

## 1. Introduction

Canine degenerative myelopathy (CDM) has been known for almost 50 years as a spontaneously occurring disease, with onset in adult dogs (around the age of 7–8 years) and characterized by neurodegenerative processes of the spinal cord. CDM was first described by Averill, D.R., in 1973, who presented the symptomatology and pathological findings in aging German Shepherd dogs [1].

CDM usually has a slow course, debuting with ataxia of the hind limbs and progressing towards severe weakness. Initially, the affected dogs stumble, knuckling appears, and subsequently the nails are worn. The symptoms usually indicate lesions of the upper motor neurons, characterized by exaggerated spinal reflexes, increased muscular tone, and proprioceptive deficits [2].

The antemortem diagnosis is based on the recognition of the clinical symptom progression, followed by a differential diagnosis in order to exclude diseases with similar symptoms. Neurological diagnosis techniques for the evaluation of the spinal cord include cerebrospinal fluid analysis, electromyography, and imaging techniques for the analysis of the spinal cord such as myelography and magnetic resonance imaging [3,4]. The differential diagnosis should take into consideration diseases with similar clinical manifestations such as: degenerative lumbosacral syndrome, spinal cord neoplasia, intervertebral disc disease, cranial cruciate ligament rupture, and hip dysplasia [3].

The postmortem exam includes the histopathological examination, which highlights lesions of primary central axonopathy restricted to the spinal cord. The myelin of the axons located in the spinal cord follows a degenerative, segmented pattern that affects all of the funiculi, which mainly involve proprioception, sensitive neurons, and somatic motor neurons. Degenerations of the neuronal perikarya were not identified [5,6]. The presumed mechanism concerns the accumulation of canine mutant *SOD1* proteins in the spinal neurons and astrocytes [7,8]. The progressive nature of CDM may be determined by the cell-to-cell propagation of *SOD1* aggregates [9]. Additionally, the neuroinflammation plays an important role in the mechanism of CDM, as microglia and astrocytes from the microenvironment of the affected spinal cord produce a set of cytokines, chemokines, and expressing adhesion molecules [10].

CDM is characterized as an autosomal recessive disease with incomplete penetrance. There are two known causal mutations. The first one was reported in 2009 in breeds such as Boxer, Rhodesian Ridgeback, German Shepherd, Pembroke Welsh Corgi, and Chesapeake Bay Retrievers and is a G to A transition (c.118G > A) in exon 2 of the gene *SOD1*, which leads to a change from glutamate to lysine in amino acid 40 of the superoxide dismutase (SOD) polypeptide chain [7]. The other mutation was reported in a Bernese Mountain Dog in 2011, and it is located in exon 1 of the same *SOD1* gene. This leads to a change in amino acid 18 from threonine to serine [11]. Additionally, in Pembroke Welsh Corgis, variations have been observed in genetic transcription mediated by haplotype SP110 (nuclear body protein), which may be partially associated with the risk of developing CDM at a younger age in Pembroke Welsh Corgi dogs homozygous for the disease-related *SOD1* mutation [12].

Currently, there is no treatment for CDM. Although there have been assumptions regarding the autoimmune nature of the disease, corticosteroid therapy did not provide any long-term improvements in the clinical progression of the disease. It was noticed that physiotherapy may slow down the disease’s progression and improve the quality of life of the affected patients [2].

A study conducted in vitro on macrophage migration inhibitory factors showed its’ potential therapeutic effect. Pathologic processes associated with the degeneration of the nervous tissue are associated with the aggregate formation of mutant protein *SOD1* E40K in the neuronal cytoplasm. The protein aggregates are formed due to protein misfolding. Macrophage migration inhibitory factor showed inhibitory effects on the misfolding of *SOD1* protein and improvement of the mutant *SOD1*-expressing motor neurons [13]. Additionally, novel oxindole compounds have shown inhibitory effects on aggregate formation of the canine mutant superoxide dismutase 1 (cSOD1 E40K) [14].

CDM exhibits several molecular and clinical characteristics that are similar to some types of human amyotrophic lateral sclerosis (ALS). These similarities include the pattern of disease progression and the distribution of lesions, which resemble those observed in the form of ALS that is dominated by upper motor neuron involvement. In addition to the loss of motor neurons, dogs with CDM exhibit several other pathological characteristics that are also observed in rodent models and in human patients with SOD1-ALS. These include injury to oligodendrocytes, resulting in demyelination, an increase in microglia expressing arginase 1 in the vicinity of motor neurons, and an increase in CB2 receptors in reactive astrocytes [15].

The aim of this study is to determine the frequency of the *SOD1.*c.118G > A mutant allele associated with CDM in breeding kennels from Romania.

## 2. Materials and Methods

### 2.1. Sample Collection

The present study was carried out between November 2021 and June 2022 and involved 230 dogs. All dogs are registered with the Romanian Kennel Club. Buccal swab samples were collected from each dog using sterile cotton swabs (Prima, Taizhou Honod Medical Co., Ltd., Zhejiang, China). The samples were collected with the consent of the dog breeders, who voluntarily participated in the study. The dogs did not exhibit any clinical signs specific to CDM when the samples were collected.

The breeding kennels were selected based on the willingness of the owners to take part in the study. Most of the kennels were from the Transylvania region of Romania, including the following counties: Cluj (19 kennels, 115 dogs), Bistrița-Năsăud (two kennels, 37 dogs), and Sibiu (one kennel, 15 dogs). In addition, kennels from the Maramureș and Oltenia regions were also included. In the Maramureș region, the kennels were located in Satu Mare (2 kennels, 23 dogs) and Maramureș (3 kennels, 25 dogs) counties, while in the Oltenia region, there was 1 kennel (15 dogs) located in Vâlcea county. The pedigree was taken into consideration when selecting the animals included in the study to certify the dog breed.

A total of 28 breeding kennels were included in the study. All of the dogs from the included kennels took part in the study. Nine of the kennels had more than one breed, as follows: 7 kennels with 2 breeds and 2 kennels with 3 breeds. The requirement for joining the Romanian Kennel Club is to have at least one dog with a certified pedigree. Under these circumstances, six of the breeding kennels had only one dog available. The number of dogs per kennel ranged between 1 and 28, and the mean number of dogs per kennel was 8.2. The ages of the dogs ranged between 2 months and 13 years at the time of sampling, with a mean of 4.4 years. Out of the total number of 230 dogs, 138 were bitches and 92 were studs.

There were a total of 26 dog breeds included in the study, as follows: Romanian Bucovina Shepherd, Romanian Mioritic Shepherd, Caucasian Shepherd, German Shepherd, Belgian Shepherd, Czechoslovakian Wolfdog, Wire Fox Terrier, Tibetan Mastiff, Saint-Bernard, Central Asia Shepherd, Labrador Retriever, Italian Cane Corso, Rottweiler, American Staffordshire Terrier, Beagle, Bull Terrier, Staffordshire Bull Terrier, Flat Coated Retriever, Dogo Argentino, Tosa, Golden Retriever, Kangal Shepherd, Bernese Mountain Dog, Shar Pei, French Bulldog, and American Bulldog.

### 2.2. DNA Extraction and Amplification of SOD1 Gene

DNA was individually extracted from all oral swabs collected from the dogs using a commercial kit (Isolate II Genomic DNA Kit, Meridian Bioscience, Newtown, OH, USA), in accordance with the manufacturer’s instructions.

The extracted DNA was tested for the presence of the canine *SOD1* gene by conventional polymerase chain reaction (PCR). PCR was carried out in a total volume of 25 μL. The reaction mixture consisted of 12.5 μL of MyTaq Red HS Mix (Meridian Bioscience, Newtown, OH, USA), 25 pM of each primer: DM_F (5′-AGTGGGCCTGTTGTGGTATC-3′) and DM_R (5′-TCTTCCCTTTCCTTTCCACA-3′) (Generi-Biotech, Hradec Králové, Czech Republic), and 4 μL of DNA template. A Bio-Rad C1000TM Thermal Cycler (Bio-Rad Laboratories, Hercules, CA, USA) was used to perform PCR amplification. Cycling conditions consisted of 5 min at 95 °C for initial denaturation, followed by 40 cycles of 40 s at 94 °C for denaturation, 30 s at 55 °C for hybridization, 1 min at 72 °C for extension, and 5 min at 72 °C for final extension. The PCR products were electrophoresed on a 2% agarose gels and stained with RedSafe Nucleic Acid Staining Solution 20,000× (iNtRON Biotechnology, Inc., Gyeonggi-do, Republic of Korea). The UV light (Bio-Rad BioDoc-ItTM Imagine System, Bio-Rad Laboratories, Hercules, CA, USA) and a 100 bp DNA ladder (Fermentas; Thermo Fisher Scientific, Waltham, MA, USA) were used for examination. The expected result was an amplicon of 292 base pairs (bp).

### 2.3. Investigation of c.118G > A Mutation in Exon 2 of the SOD1 Gene Using the Restriction Fragment Length Polymorphism (RFLP)

The PCR products were digested using AcuI enzyme (New England Biolabs, Ipswich, MA, USA) using the following mix: 2.5 µL digestion buffer, 11.5 µL ultrapure water, 1 µL AcuI enzyme, and 10 µL PCR product. The samples were incubated at 37 °C for 6 h, followed by the inactivation of the enzyme at 65 °C for 20 min.

After digestion, the products were migrated in a 3% agarose gel stained with RedSafe Nucleic Acid Staining Solution 20000× (iNtRON Biotechnology, Inc., Gyeonggi-do, Republic of Korea), examined under UV light (Bio-Rad BioDoc-ItTM Imagine System, Bio-Rad Laboratories, Hercules, CA, USA), and compared with a 100 bp DNA ladder (Fermentas; Thermo Fisher Scientific, Waltham, MA, USA). The wild-type homozygotes (G/G) were expected to have two amplicons (230 bp and 62 bp), the heterozygotes (G/A) were expected to have three amplicons (292 bp, 230 bp, and 62 bp), and homozygotes with the mutation associated with CDM (AA) were expected to have one amplicon of 292 bp (Figure 1).

### 2.4. Statistical Analysis

The allele frequency was determined, and the Hardy–Weinberg testing was done using HW_TEST software [16,17]. The variations between the measured and expected differences were considered statistically significant at *p* < 0.05, while data at *p* ≥ 0.05 were considered within the Hardy–Weinberg equilibrium. The Pearson correlation coefficient was applied using SPSS Statistics Version 26 (IBM Corp., Armonk, NY, USA) to compare the mutant allele frequency results obtained in this study with the results of another study. The same software was used to generate the scatter plot.

## 3. Results

The genotyping of the breeds revealed that out of the total of 230 dogs, 204 were homozygous for the wild-type allele (G/G), 16 were heterozygous (A/G), and 10 were homozygous for the mutant allele (A/A). The mutant allele (A) was identified in 6 of the total 26 breeds. The heterozygous individuals belonged to the German Shepherd, Belgian Shepherd, Romanian Mioritic Shepherd, and Czechoslovakian Wolfdog breeds, while the mutant homozygous individuals were part of the Wire Fox Terrier, German Shepherd, and Rottweiler breeds. The distribution of the genotypes amongst the dog breeds is shown in Table 1.

The frequency of the mutant allele (A) varied among the tested population and different dog breeds. Specifically, it was found to be 0.0783 in the entire tested population, 1 in Wire Fox Terrier, 0.2500 in Romanian Mioritic Shepherd, 0.2000 in German Shepherd, 0.0833 in Rottweiler, and 0.0313 in Belgian Shepherd.

Whereas the genotypic frequencies for the Belgian Shepherd, German Shepherd, and Romanian Mioritic Shepherd were within the Hardy–Weinberg equilibrium, a departure was observed for Rottweiler (Table 2).

## 4. Discussion

The current study represents a screening of dogs belonging to different breeding kennels in Romania, and as per our knowledge, for the certified Romanian dog breeds Romanian Mioritic Shepherd and Romanian Bucovina Shepherd, this was the first documentation of CDM screening. This was also the first CDM screening for the Caucasian Shepherd. The mutant allele frequency (A) within the entire tested population (*n* = 230) was 0.0783, with 204 dogs homozygous for the wild-type allele (G/G), 16 heterozygous (A/G), and 10 homozygous for the mutant allele (A/A). The dogs included in the study did not exhibit any clinical signs specific to CDM.

The Romanian Bucovina Shepherd breed standard was published on 14 May 2018 by the International Canine Federation (ICF). The dogs belonging to this breed are used for herding and guarding, having their origin in the Bucovina region, in the north-eastern region of Romania [18]. Based on mitochondrial DNA sequences, the closest breeds are the Terra Nova and the Tibetan Mastiff, followed at a greater genetic distance by the Bobtail, German Shepherd, and Australian Shepherd breeds [19]. So far, there has been no research on the occurrence of the mutation linked to degenerative myelopathy in this particular breed. The mutated gene was not detected in the tested individuals from the Romanian Bucovina Shepherd breed.

The mutant allele was identified in the Romanian Mioritic Shepherd breed. As per the ICF’s description, the breed was selected for its practicality, as it originated from a naturally existing breed found in the Carpathian Mountains [20]. The Treeing Walker Coonhound, Shetland Sheepdog, and Saint-Bernard breeds were identified as the breeds most closely related to the Romanian Mioritic Shepherd breed, based on mitochondrial DNA sequences [19]. The mutant allele frequency was 0.2500.

The study also included eight Caucasian Shepherd dogs with no previous documentation of testing for CDM. This breed has its origins in the Caucasus Mountains and the steppe regions of southern Russia, where it was used for guarding and herding. Breed selection began in the 1920s in the USSR [21]. We did not find the mutant allele in the Caucasian Shepherd breed.

A large-scale study that included 33,746 dogs was carried out in 2014 to determine the distribution of the CDM-associated *SOD1:*c.118G > A mutation. Almost half (49%) of the dogs included in the study were homozygous for normal alleles (G/G), while 27% were heterozygous (G/A) and 24% were homozygous (A/A) for the *SOD1*:c.118G > A mutation. The study included dogs from 222 breeds, and 1368 dogs were mixed breeds. The mutant allele (A) was identified in at least one individual in 124 breeds. Dog breeds in which all individuals were homozygous for normal alleles were: Weimaraner, Great Dane, Field Spaniel, Basenji, Akita, English Cocker Spaniel, Bernese Mountain Dog, Bouvier des Flandres, and Dachshund. The same study estimated that 60% of individuals homozygous for the mutant allele developed clinical signs specific to CDM, and heterozygous individuals had a relatively similar risk of developing clinical signs as individuals homozygous for normal alleles. Estimates were made based on surveying the owners of 512 dogs [22].

The mutant allele frequency obtained in the present study was compared with the overlapping breeds that were tested by Zeng et al. [22]. The results of both studies are presented in Appendix A. A Pearson’s correlation was estimated, and the result showed a strong, positive correlation (*r* = 0.860, N = 21, *p* < 0.001) between our results and the results obtained by Zeng et al. [22], as illustrated in Figure 2.

The aforementioned study [22] is currently the largest and most representative of the epidemiology of degenerative myelopathy. Compared to our results (0.0783), it identified the frequency of the *SOD1:*c.118A allele as 0.37 in the entire population. These differences can be attributed to the fact that the study also included mixed-breed dogs (in which the frequency of the mutant allele was 48%) and other dog breeds with a high frequency of the mutant allele. Moreover, the study included samples from dogs with neurological symptoms, which increased the probability of identifying individuals with the mutation.

For Rottweilers, the mutant allele frequency determined by Zeng et al. [22] was 0.03, while our result was 0.0833. The statistical analysis of our results indicated a departure from the Hardy-Weinberg equilibrium. This may be caused by a variety of factors, including purifying selection, genotyping error, copy number variation, inbreeding, and population substructure [23].

For Wire Fox Terriers, the mutant allele frequency was 1, with all individuals being homozygous for the mutant allele. The Wire Fox Terrier breed was highlighted by Zeng et al. [22] as the breed with the highest mutant allele frequency among the tested population in their study (0.94 for 78 tested individuals). However, the prevalence of Wire Fox Terriers exhibiting symptoms of CDM is currently unknown.

The Czechoslovakian Wolfdog breed is the result of an interbreeding program between a German Shepherds and Carpathian wolves [24]. Regarding CDM, the individual that was tested in the present study was heterozygous. As stated in Appendix A, Zeng et al. [22] identified the mutant allele frequency as 0.3400. Another study conducted in Slovakia on 54 Czechoslovakian Wolfdogs showed an allele frequency of 0.2500 [25]. This relatively high mutant allele frequency may be attributed to the relatedness of the Czechoslovakian Wolfdog to the German Shepherd.

CDM is well documented in the German Shepherd breed, as it is the first breed in which the condition was described. In the case of the present study, the mutant allele frequency was 0.20 for the German Shepherd breed. A study conducted in Brazil estimated the frequency of the mutant allele to be 0.12 in the German Shepherd breed [16]. In the United Kingdom, the mutant allele frequency in a reference population of German Shepherds was estimated to be 0.38 [26], while in Japan it was determined to be 0.22 [27]. In the USA, the mutant allele frequency was 0.36 [22] and 0.33 [28] in the German Shepherd population, while in Poland it was 0.18 and in Israel it was 0.17 [28]. In Mexico, the mutant allele frequency in the tested German Shepherd population was 0.13 [29]. One additional study, which included German Shepherd samples from Germany, Belgium, and the Netherlands, identified an allele frequency of 0.15 [30].

For Belgian Shepherd, our results showed a mutant allele frequency of 0.0313. For the same breed, the frequency of the mutant allele was 0.06 in the study conducted by Zeng et al. [22], while a study conducted in Greece determined it to be 0.17 [31]. In Mexico, the mutant allele frequency in the tested Belgian Shepherd population was 0.14 [29].

A limitation of the present study is that the relationship between individuals within the same kennel was not taken into consideration when analyzing the data. Hence, estimates of the frequency of the mutated allele may have been biased upward due to identity-by-descent relationships. In addition, in some instances, the number of individuals sampled was too small to accurately estimate the mutant allele frequency of the breeds.

## 5. Conclusions

The present study presents important data related to the occurrence of the *SOD1:*c.118G > A mutation in 28 kennel dogs from Romania. According to our knowledge, this study reports the first screening of dogs of the Romanian Bucovina Shepherd, Caucasian Shepherd, and Romanian Mioritic Shepherd breeds for CDM.

The rate of success of DNA isolation and amplification (PCR-RFLP) was 100%, proving that oral swabs represent good sources for genetic testing of CDM. Of 230 tested dogs, 204 were homozygous for the wild-type allele (G/G), 16 were heterozygous (A/G), and 10 were homozygous for the mutant allele (A/A). The mutant allele (A) was identified in 6 of the 26 tested breeds. The mutant allele frequency (A) associated with CDM within the 230 swab samples was 0.0783. Segregation of the results by breed indicated a frequency of 1 in Wire Fox Terriers, 0.2500 in Romanian Mioritic Shepherd, 0.2000 in German Shepherd, 0.0833 in Rottweilers, and 0.0313 in Belgian Shepherd. The *SOD1*:c118G > A mutation was identified for the first time in Romanian Mioritic Shepherds. However, the mutation was not found in Romanian Bucovina Shepherds or in Caucasian Shepherds.

Considering that CDM is an autosomal recessive disorder with incomplete penetrance, genetic testing is important for genetic prophylaxis in order to avoid the emergence of dogs homozygous for the *SOD1*:c118G > A risk allele, as these homozygous dogs will be at very high risk of developing CDM.

Genetic testing is also an important tool to be used for the differential diagnosis of CMD and similar neurological pathologies, such as: degenerative lumbosacral syndrome, spinal cord neoplasia, intervertebral disc disease, cranial cruciate ligament rupture, and hip dysplasia.

## Figures and Tables

**Figure 1 animals-13-01403-f001:**
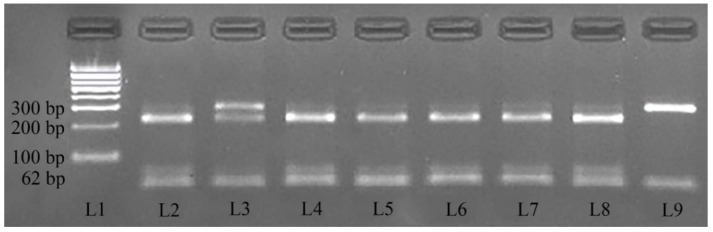
PCR-RFLP electrophoresis gel results highlighting the three genotypes. Legend: L1 corresponds to size standard (100-bp DNA ladder); samples L2, L4, L5, L6, L7, and L8 correspond to genotype G/G (230 bp and 62 bp); sample L3 corresponds to genotype A/G (292 bp, 230 bp, and 62 bp); sample L9 corresponds to genotype AA (292 bp).

**Figure 2 animals-13-01403-f002:**
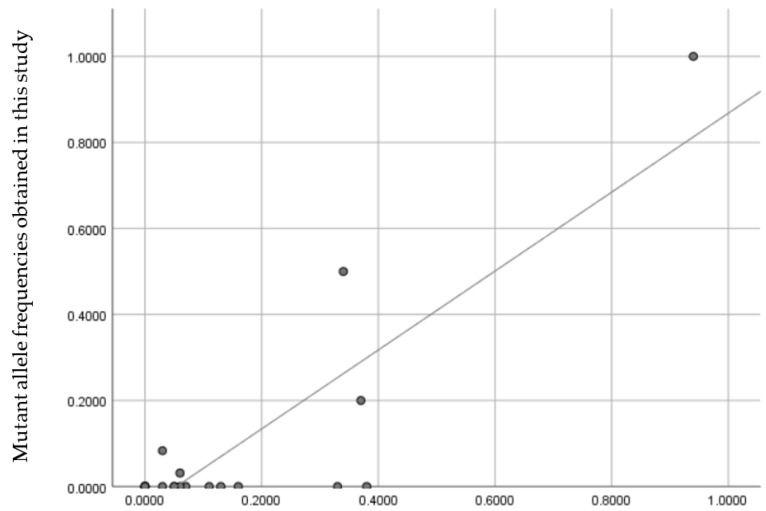
Scatter plot showing the correlation between the mutant allele frequency obtained in our study and the mutant allele frequency obtained by Zeng et al. 2014 [22]. Each point represents a dog breed. The source data for the figure is presented in Appendix A.

**Table 1 animals-13-01403-t001:** Genotype distribution among dog breeds.

			Genotype			
Breed	Number of Kennels per Breed	Number of Tested Dogs	G/G	G/A	A/A	Mutant Allele Frequency
Wire Fox Terrier	2	7	-	-	7	1.000
Romanian Mioritic Shepherd	1	4	2	2	-	0.2500
German Shepherd	3	40	26	12	2	0.2000
Rottweiler	2	12	11	-	1	0.0833
Belgian Shepherd	2	16	15	1	-	0.0313
Czechoslovakian Wolfdog	1	1	-	1	-	0.5000
Romanian Bucovina Shepherd	2	25	25	-	-	n/a
Caucasian Shepherd	1	8	8	-	-	n/a
Tibetan Mastiff	1	5	5	-	-	n/a
Saint-Bernard	1	20	20	-	-	n/a
Central Asia Shepherd	2	17	17	-	-	n/a
Labrador Retriever	2	14	14	-	-	n/a
Italian Cane Corso	3	13	13	-	-	n/a
American Staffordshire Terrier	1	11	11	-	-	n/a
Beagle	1	10	10	-	-	n/a
Bull Terrier	1	7	7	-	-	n/a
Staffordshire Bull Terrier	2	6	6	-	-	n/a
Flat Coated Retriever	1	3	3	-	-	n/a
Dogo Argentino	1	2	2	-	-	n/a
Tosa	1	2	2	-	-	n/a
Golden Retriever	1	2	2	-	-	n/a
Kangal Shepherd	1	1	1	-	-	n/a
Bernese Mountain Dog	1	1	1	-	-	n/a
Shar Pei	1	1	1	-	-	n/a
American Bulldog	1	1	1	-	-	n/a
French Bulldog	1	1	1	-	-	n/a
Total	n/a *	230	204	16	10	0.0783

Legend: n/a—not applicable; *—a total of 26 dog breeds belonging to 28 breeding kennels were included in the study. The total number of kennels involved in the study does not correspond with the number of kennels per breed, as nine of the breeding kennels included more than one breed.

**Table 2 animals-13-01403-t002:** Hardy–Weinberg equilibrium test results.

	Observed Genotype Frequency			Expected Genotype Frequency			
Breed	G/G	G/A	A/A	G/G	G/A	A/A	Chi-squared test analysis
Belgian Shepherd	0.937	0.063	0	0.938	0.061	0.001	*p =* 0.8973
German Shepherd	0.650	0.300	0.050	0.640	0.320	0.040	*p* = 0.6926
Rottweiler	0.917	0	0.083	0.840	0.153	0.007	*p =* 0.0005
Romanian Mioritic Shepherd	0.500	0.500	0	0.562	0.375	0.063	*p =* 0.5050

## Data Availability

All the results of the study are presented within the manuscript.

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
