# Peer review of "Molecular Surveillance of Canine Degenerative Myelopathy in Breeding Kennels from Romania"

_animals, 2023, doi:10.3390/ani13081403_

Round 1
Reviewer 1 Report
Review of the paper titled “Molecular surveillance of Canine Degenerative Myelopathy in breeding kennels”
General comments
Aim of the reviewed manuscript was to perform a first screening of the mutation associated with canine degenerative myelopathy in dogs bred in Romania. The authors presented a straightforward and scientific-sound work. The manuscript is well written and build and, in my opinion, has just a single issue that should be addressed before publication, as described below.
In the Materials and Methods section the authors described sample collection and specified the breeds involved. However, there is nothing regarding the animals selection: authors should describe the criteria used in dogs selection.
I have a few questions to be addressed in the manuscript.
- Did each breeder selected which dogs to select?
- How many kennels in total and how many kennels/breed were involved?
- Was it a randomized selection among the available ones?
- Were the pedigree information taken into account when selecting animals?
- The selection of related dogs could have biased the representativeness of the sample over the population, increasing the frequency of the mutated allele because identity-by-descent. Is it possible to add some population parameters to describe the sampled populations?
- Furthermore, there are breeds highly represented in the study (e.g., GSD), while others were represented by a single dog. Could you justify this unbalanced situation?
In addition to these questions, at line 176 the authors mentioned determining HWE for Wire Fox Terrier breed, please double-check.
English and style could be improved.
In conclusion, I believe this paper should be reconsidered after MAJOR REVISIONS.
Author Response
Dear Reviewer,
We appreciate all of the constructive criticisms, useful comments and thoughtful suggestions. All substantive points and suggestions have been carefully considered during revision of the original manuscript and implemented (please see below).
A detailed response to all comments from the reviewers is outlined in the accompanying letter indicated by sentences starting with AU:. The original paper has been extensively revised in accordance with the comments and recommendations arising from the peer-review process.
We hope that the revisions would allow the manuscript to be considered acceptable for publication.
With our best regards,
Dr. Anamaria Ioana Paștiu
Reviewer #1
Review of the paper titled “Molecular surveillance of Canine Degenerative Myelopathy in breeding kennels”
General comments
Aim of the reviewed manuscript was to perform a first screening of the mutation associated with canine degenerative myelopathy in dogs bred in Romania. The authors presented a straightforward and scientific-sound work. The manuscript is well written and build and, in my opinion, has just a single issue that should be addressed before publication, as described below.
In the Materials and Methods section the authors described sample collection and specified the breeds involved. However, there is nothing regarding the animals selection: authors should describe the criteria used in dogs selection.
I have a few questions to be addressed in the manuscript.
- Did each breeder selected which dogs to select?
AU: The breeders did not select the dogs which were involved in the study. All of the dogs from the included kennels took part in the study. We have updated the Materials and Methods section to include this clarifying aspect.
- How many kennels in total and how many kennels/breed were involved?
AU: The total number of kennels included in the study was 28. The number of kennels per breed is now included in Table 1.
- Was it a randomized selection among the available ones?
AU: The breeding kennels were selected based on the willingness of the owners to take part in our study. Also, the kennels selected were mostly from the Transylvania region of Romania. We have added this clarification in the Materials and Methods section.
- Were the pedigree information taken into account when selecting animals?
AU: The pedigree was taken into consideration when selecting the animals included in the study only as a proof of certification of the dog breed. Other information from the pedigree (eg. ancestry) was not taken into consideration when selecting the individuals and interpreting the data. The Materials and Methods section now contains this clarification.
- The selection of related dogs could have biased the representativeness of the sample over the population, increasing the frequency of the mutated allele because identity-by-descent. Is it possible to add some population parameters to describe the sampled populations?
AU: Unfortunately, we do not have any data on the relationship of the sampled individuals within the breeding kennels (eg. parents and offsprings). We have added this as a limitation at the end of the discussions section.
- Furthermore, there are breeds highly represented in the study (e.g., GSD), while others were represented by a single dog. Could you justify this unbalanced situation?
AU: As mentioned above, the breeding kennels were selected based on the owners’ willingness to take part in the study. The condition for the establishment of a breeding kennel is to have at least one dog with a pedigree document to certify the breed. Also, we assume that the number of dogs per breed is also related to the popularity of each breed. Additionally, some of the GSD breed owners were aware of the breed susceptibility to CDM, hence they were more opened to take part in the study. We have updated the Materials and Methods section to include this clarification.
In addition to these questions, at line 176 the authors mentioned determining HWE for Wire Fox Terrier breed, please double-check.
AU: Thank you for pointing out this mistake. We have corrected accordingly.
English and style could be improved.
AU: Thank you for your comment. We have taken steps to improve the English and style in our manuscript.

Reviewer 2 Report
In this study, the authors present a survey on the occurrence of the Canine Degenerative Myelopathy (CDM)-linked mutation SOD1:c118G>A in a number of dogs from Kennels existing in Romania, as well as in specific Romanian dog breeds that had not been previously evaluated for this mutation. Although mainly of descriptive nature, the manuscript is well written, clear and straight-forward to understand. However, I have some suggestions I believe will improve the manuscript.
Major comments:
1) I would suggest to add an additional column with the SOD1:c118G>A risk allele frequency in the different breeds in Table 1. This will be very helpful in order to understand the risk mutation distribution in the different breeds.
2) I would suggest to extract the SOD1:c118G>A risk allele frequency in all the overlapping breeds between this study and the study by Zeng et al., 2014 (reference number 21 in the current manuscript), and to run a correlation test. It would be also helpful and informative to add a correlation plot showing how the frequency in the two studies distribute and correlate. After obtaining the outcome of the correlation test (i.e. high or low), it would be also good to discuss the reasons behind this result.
3) The authors found that the frequency of the SOD1:c118G>A risk allele was 1 in the Wire Fox Terrier (WFT) dogs. They also stated that no dogs had CDM clinical signs. I understand that what I'm suggesting here is probably out of the scope of the manuscript, but I would be curious to genotype the SP110 haplotype described in Ivansson et al., 2016 (reference number 12 in the current manuscript) in the WFT dogs and see whether these dogs carry the expected haplotype. Simplistically (maybe), the detection of the expected haplotype would mean that SP110 is potentially a modifier also in WFT. Conversely, the detection of the non-expected haplotype would suggest the existence of additional modifier loci in WFT. Does this make sense? Any comment?
4) In the discussion, the authors end three paragraphs (first paragraph: lines 188-198; second paragraph: lines 199-205; third paragraph: lines 206-211) repeating the same conclusion, namely that the study sample size is small and their dataset might not be representative. This is redundant and does not help the flow of the text. I would restructure this part of the discussion, so that there is no such a repetition.
5) In the paragraph included between lines 233 and 236, the authors expanded on the results of Hardy-Weinberg equilibrium (HWE) in Rottweilers by listing a number of general reasons for HWE departures. Could the authors speculate on the most likely reason(s) for the result they had gotten for Rottweilers in their study?
6) As far as I noticed, in at least three instances (end of the simple summary, end of the abstract, and conclusions) the authors wrote that genetic testing is important for prevention of mutation/disease spreading within dog populations. The point is that, also given that CDM is autosomal recessive with incomplete penetrance (as the authors stated), genetic testing is important to avoid the generation of dogs homozygous for the SOD1:c118G>A risk allele, because these homozygous dogs will be at very high risk of developing CDM. Once homozygous A/A dogs are not produced, then also the spread of the disease is also diminished. Genetic testing is also important to keep the frequency of the SOD1:c118G>A risk allele very low in the population, not to completely remove it, which would cause loss of genetic variability and inbreeding. So, I would rephrase in all those three instances I pointed out (and possibly others I've failed to notice).
Minor comments:
1) I suggest that the title will be changed to "Molecular Surveillance of Canine Degenerative Myelopathy in Breeding Kennels from Romania". This better reflects the study, which is focused on dogs only from this country.
2) Line 14/15: I would say "...associated mutation in dogs from different breeding kennels in Romania."
3) Line18: The word "affected" is misleading here. Given that the authors say that the dogs didn't exhibit any clinical sign of the disease at sampling, I would replace "affected" with "homozygous for the risk allele".
4) Line 21: I would say "...tool used for the confirmation of affected individuals and prevention...", because being homozygous for the risk allele does not always mean having the disease, which needs to be clinically diagnosed.
5) Line 44: Replace "is known" with "has been known".
6) Line 74: Replace "of gene SOD1" with "of the gene SOD1".
7) Line 80: delete the word "period", I feel that it does not add anything.
8) Line 80-82: I would say "Variations in genetic transcription mediated by the SP110 (nuclear body protein) haplotype observed in this dog breed may be partially associated with the risk of developing CDM at a younger age in PWC dogs homozygous for the disease-related SOD1 mutation [12]".
9) Line 95: I would say "...is to determine the frequency of the SOD1.c118G>A mutant allele associated with CDM in...".
10) Line 111-112: I would say "The age of the dogs ranged between 2 and 13 months at the time of sampling. 138 dogs were females and...".
11) Line 116: replace "from dogs" with "from the dogs".
12)Line 133: Remove ", corresponding to canine SOD1 gene". It sounds like you amplified the whole gene, which is obviously not the case.
13) Line 161: Replace "204 were" with "204 dogs were"
14) Line 183-184: Replace "breeding kennels, " with "breeding kennels in Romania, "
Author Response
Dear Reviewer,
We appreciate all of the constructive criticisms, useful comments and thoughtful suggestions. All substantive points and suggestions have been carefully considered during revision of the original manuscript and implemented (please see below).
A detailed response to all comments from the reviewers is outlined in the accompanying letter indicated by sentences starting with AU:. The original paper has been extensively revised in accordance with the comments and recommendations arising from the peer-review process.
We hope that the revisions would allow the manuscript to be considered acceptable for publication.
With our best regards,
Dr. Anamaria Ioana Paștiu
Reviewer #2
In this study, the authors present a survey on the occurrence of the Canine Degenerative Myelopathy (CDM)-linked mutation SOD1:c118G>A in a number of dogs from Kennels existing in Romania, as well as in specific Romanian dog breeds that had not been previously evaluated for this mutation. Although mainly of descriptive nature, the manuscript is well written, clear and straight-forward to understand. However, I have some suggestions I believe will improve the manuscript.
Major comments:
1) I would suggest to add an additional column with the SOD1:c118G>A risk allele frequency in the different breeds in Table 1. This will be very helpful in order to understand the risk mutation distribution in the different breeds.
AU: The mutant allele frequency was removed from Table 2 and included in Table 1.
2) I would suggest to extract the SOD1:c118G>A risk allele frequency in all the overlapping breeds between this study and the study by Zeng et al., 2014 (reference number 21 in the current manuscript), and to run a correlation test. It would be also helpful and informative to add a correlation plot showing how the frequency in the two studies distribute and correlate. After obtaining the outcome of the correlation test (i.e. high or low), it would be also good to discuss the reasons behind this result.
AU: We have run a Pearson correlation on the results and added a short discussion. The results distribution are now included in Figure 2.
3) The authors found that the frequency of the SOD1:c118G>A risk allele was 1 in the Wire Fox Terrier (WFT) dogs. They also stated that no dogs had CDM clinical signs. I understand that what I'm suggesting here is probably out of the scope of the manuscript, but I would be curious to genotype the SP110 haplotype described in Ivansson et al., 2016 (reference number 12 in the current manuscript) in the WFT dogs and see whether these dogs carry the expected haplotype. Simplistically (maybe), the detection of the expected haplotype would mean that SP110 is potentially a modifier also in WFT. Conversely, the detection of the non-expected haplotype would suggest the existence of additional modifier loci in WFT. Does this make sense? Any comment?
AU: Indeed, it would be interesting to genotype the SP110 haplotype and determine its potential as a modifier in regards to CDM risk and age of onset. However, considering the tight timelines, we are currently unable to address this suggestion.
Ivansson et al., (2016) found a positive correlation in Pembroke Welsh Corgis (PWC) of the haplotype presence and the increase probability of developing CDM symptoms and an earlier onset age. Conversely, the Boxers were found to be genetically similar to PWC across the SP110 locus, but unlikely to have any changes in the risk of CDM. This indicates, as you said, the possibility of additional loci acting as modifiers.
This would be a great topic for research into CDM, and a new approach towards the particular situation of WFT.
4) In the discussion, the authors end three paragraphs (first paragraph: lines 188-198; second paragraph: lines 199-205; third paragraph: lines 206-211) repeating the same conclusion, namely that the study sample size is small and their dataset might not be representative. This is redundant and does not help the flow of the text. I would restructure this part of the discussion, so that there is no such a repetition.
AU: We have removed these paragraphs and included a segment at the end of the discussions section that now mentions the limitations.
5) In the paragraph included between lines 233 and 236, the authors expanded on the results of Hardy-Weinberg equilibrium (HWE) in Rottweilers by listing a number of general reasons for HWE departures. Could the authors speculate on the most likely reason(s) for the result they had gotten for Rottweilers in their study?
AU: The departure from the Hardy-Weinberg equilibrium may be related to the population substructure, considering the fact that the Rottweilers were part of two different kennels. This is now incorporated in the discussions section.
6) As far as I noticed, in at least three instances (end of the simple summary, end of the abstract, and conclusions) the authors wrote that genetic testing is important for prevention of mutation/disease spreading within dog populations. The point is that, also given that CDM is autosomal recessive with incomplete penetrance (as the authors stated), genetic testing is important to avoid the generation of dogs homozygous for the SOD1:c118G>A risk allele, because these homozygous dogs will be at very high risk of developing CDM. Once homozygous A/A dogs are not produced, then also the spread of the disease is also diminished. Genetic testing is also important to keep the frequency of the SOD1:c118G>A risk allele very low in the population, not to completely remove it, which would cause loss of genetic variability and inbreeding. So, I would rephrase in all those three instances I pointed out (and possibly others I've failed to notice).
AU: Thank you for the suggestions. We have clarified throughout the text accordingly.
Minor comments:
1) I suggest that the title will be changed to "Molecular Surveillance of Canine Degenerative Myelopathy in Breeding Kennels from Romania". This better reflects the study, which is focused on dogs only from this country.
2) Line 14/15: I would say "...associated mutation in dogs from different breeding kennels in Romania."
3) Line18: The word "affected" is misleading here. Given that the authors say that the dogs didn't exhibit any clinical sign of the disease at sampling, I would replace "affected" with "homozygous for the risk allele".
4) Line 21: I would say "...tool used for the confirmation of affected individuals and prevention...", because being homozygous for the risk allele does not always mean having the disease, which needs to be clinically diagnosed.
5) Line 44: Replace "is known" with "has been known".
6) Line 74: Replace "of gene SOD1" with "of the gene SOD1".
7) Line 80: delete the word "period", I feel that it does not add anything.
8) Line 80-82: I would say "Variations in genetic transcription mediated by the SP110 (nuclear body protein) haplotype observed in this dog breed may be partially associated with the risk of developing CDM at a younger age in PWC dogs homozygous for the disease-related SOD1 mutation [12]".
9) Line 95: I would say "...is to determine the frequency of the SOD1.c118G>A mutant allele associated with CDM in...".
10) Line 111-112: I would say "The age of the dogs ranged between 2 and 13 months at the time of sampling. 138 dogs were females and...".
11) Line 116: replace "from dogs" with "from the dogs".
12) Line 133: Remove ", corresponding to canine SOD1 gene". It sounds like you amplified the whole gene, which is obviously not the case.
13) Line 161: Replace "204 were" with "204 dogs were"
14) Line 183-184: Replace "breeding kennels, " with "breeding kennels in Romania, "
AU: All of the above-mentioned minor comments have been addressed throughout the text as per the reviewer’s suggestion.

Round 2
Reviewer 1 Report
Review R2 of the paper titled “Molecular surveillance of Canine Degenerative Myelopathy in breeding kennels”
General comments
I would like to thank the authors for their effort in answering to my comments in the previous round of this review process. However, I still have comments that need to be addressed before considering publishing.
Specific comments by sections
Materials and methods
L118 – Here the authors wrote that 28 is the number of breeding kennels involved in the study. However, in the legend of Table 1 they specify that this number “does not correspond to (and not with) the total number of kennels involved […]”. It is not clear how many “real” kennels were included. In M&M section the authors should specify the real total number of breeding kennel included in the study, and specify how many of them had more than one breed.
L122 – Please specify the mean number of dogs/kennel in addition to the range.
L129-130 – Please specify the mean age of the dogs in addition to the range.
L131 – When referring to breeding dogs, studs/bitches are preferred rather than males/females.
L142 – “was used to performed the PCR […]” -> “was use to perform PCR […]”.
Results
L192-193 – Please check the format of the legend and the usage of superscripts/indexes.
Discussion
L267 – Figure 2 is too large for the page and has been cropped. The last visible point is the Golden Retriever breed. Please fix the issue.
L248 – Figure 2 legend is quite concise. Probably a few more info are needed for the reader to understand the figure at a glance.
L261-263 – The authors’ hypothesis of a substructure in their Rottweiler sampled population is interesting. However, it is difficult to verify it with the available data. I suggest either expanding these lines, maybe trying analysing the animals from the two kennels separately (the Rottweiler sample, however, is a bit too small for this type of analysis) and therefore giving the reader something more than a mere speculation, or removing the lines themselves.
L269-274 – Interesting as it is, I believe that the history of this breed is not really necessary nor relevant here, differently from the one for Romanian breeds at the beginning of the Discussion section which fits smoothly. Please consider removing these lines.
English and style check, especially of the newly added parts, is recommended.
In conclusion, I believe this paper should be reconsidered after MAJOR REVISIONS.
Author Response
Dear Reviewer,
We appreciate all of the constructive criticisms, useful comments and thoughtful suggestions. All substantive points and suggestions have been carefully considered during revision of the original manuscript and implemented (please see below).
A detailed response to all comments is outlined in the accompanying letter indicated by sentences starting with AU:. The original paper has been extensively revised in accordance with the comments and recommendations arising from the peer-review process, and as a result is much improved.
We hope that the revisions would allow the manuscript to be considered acceptable for publication.
With our best regards,
Dr. Anamaria Ioana Paștiu
Reviewer #1
Review R2 of the paper titled “Molecular surveillance of Canine Degenerative Myelopathy in breeding kennels”
General comments
I would like to thank the authors for their effort in answering to my comments in the previous round of this review process. However, I still have comments that need to be addressed before considering publishing.
Specific comments by sections
Materials and methods
L118 – Here the authors wrote that 28 is the number of breeding kennels involved in the study. However, in the legend of Table 1 they specify that this number “does not correspond to (and not with) the total number of kennels involved […]”. It is not clear how many “real” kennels were included. In M&M section the authors should specify the real total number of breeding kennel included in the study, and specify how many of them had more than one breed.
AU: The total number of kennels was 28, as specified in the M&M section. Nine of the kennels had more than one breed, as follows: 7 kennels with 2 breeds and 2 kennels with 3 breeds. We have added the clarification in the legend of Table 1.
L122 – Please specify the mean number of dogs/kennel in addition to the range.
AU: The mean number of dogs per kennel was 8.2. We have added this clarification in the M&M section.
L129-130 – Please specify the mean age of the dogs in addition to the range.
AU: The mean age of the dogs was 4.4 years. We have added this clarification in the M&M section.
L131 – When referring to breeding dogs, studs/bitches are preferred rather than males/females.
AU: We have corrected accordingly.
L142 – “was used to performed the PCR […]” -> “was use to perform PCR […]”.
AU: Done.
Results
L192-193 – Please check the format of the legend and the usage of superscripts/indexes.
AU: We have corrected the legend and table accordingly.
Discussion
L267 – Figure 2 is too large for the page and has been cropped. The last visible point is the Golden Retriever breed. Please fix the issue.
AU: Following the suggestion of the other reviewer, we have replaced the current figure with a scatter plot that illustrates the correlation of the allele frequencies obtained in our study with the ones obtained by Zeng et al. [22]
L248 – Figure 2 legend is quite concise. Probably a few more info are needed for the reader to understand the figure at a glance.
AU: We have modified in accordance to the new figure.
L261-263 – The authors’ hypothesis of a substructure in their Rottweiler sampled population is interesting. However, it is difficult to verify it with the available data. I suggest either expanding these lines, maybe trying analysing the animals from the two kennels separately (the Rottweiler sample, however, is a bit too small for this type of analysis) and therefore giving the reader something more than a mere speculation, or removing the lines themselves.
AU: We agree with you on this matter. This speculation was added following the other reviewer’s suggestion. Considering that our data is insufficient to support it, we removed it.
L269-274 – Interesting as it is, I believe that the history of this breed is not really necessary nor relevant here, differently from the one for Romanian breeds at the beginning of the Discussion section which fits smoothly. Please consider removing these lines.
AU: We have amended accordingly and only included one short statement which is relevant.
Thank you!
Reviewer 2 Report
While I thank the authors for taking the time to answer my questions and discussion points, as well as to follow my suggestions, I still have some requests that would need to be taken into consideration and addressed.
MAJOR
1) In my previous comment regarding the importance of genetic testing, I was trying to very simplistically summarise the whole picture, also using at times a colloquial tone. While I suggested to "rephrase" sentences, it seems that the authors just directly copied what I wrote and pasted it into the text, especially in the Conclusions section. This is not advisable, especially because some messages might be misinterpreted when other details are lacking (e.g., population and evolutionary genetics theory, additional info on the heterozygous dogs for the SOD1:c118G>A risk allele and their prevalence of CDM, etc.).
The most important concept to stress here is that fact that breeders should avoid the generation of homozygous A/A dogs, given that these are at very high risk of developing CDM. So I would change as follows:
-Line 37-38: I would say "in order to avoid the emergence of dogs homozygous for the SOD1:c118G>A risk allele."
-Line 309-315: I would say "Genetic testing is important to avoid the emergence of dogs homozygous for the SOD1:c118G>A risk allele, as these homozygous dogs will be at very high risk of developing CDM."
2) It seems that the authors misinterpreted my suggestions on the additional figure representing a correlation plot of the risk allele frequency in all the overlapping (in common) breeds between this study and the study by Zeng et al., 2014.
I would suggest to change Figure 2 for two reasons: a) the current figure is cut, so only one part is visible, and this is not acceptable. b) the current figure is not at all a correlation plot. A correlation plot is a scatter plot, in which the x axis represents in this case the frequency of the risk allele in one study, and the y axis the frequency of the risk allele in the other study. Each overlapping breed then appears as a single point on the graph. Ideally, the plot also shows a regression line that describe the trend of the data, in other words the correlation. I hope it is clear now.
MINOR
1) Line 46-47: I think "CDM was first described in a German Shepherd Dog in 1973" does make little sense. Shouldn't it be "CDM was first described in the German Shepherd Dog in 1973"?
Author Response
Dear Reviewer,
We appreciate all of the constructive criticisms, useful comments and thoughtful suggestions. All substantive points and suggestions have been carefully considered during revision of the original manuscript and implemented (please see below).
A detailed response to all comments is outlined in the accompanying letter indicated by sentences starting with AU:. The original paper has been extensively revised in accordance with the comments and recommendations arising from the peer-review process, and as a result is much improved.
We hope that the revisions would allow the manuscript to be considered acceptable for publication.
With our best regards,
Dr. Anamaria Ioana Paștiu
Reviewer #2
While I thank the authors for taking the time to answer my questions and discussion points, as well as to follow my suggestions, I still have some requests that would need to be taken into consideration and addressed.
MAJOR
1) In my previous comment regarding the importance of genetic testing, I was trying to very simplistically summarise the whole picture, also using at times a colloquial tone. While I suggested to "rephrase" sentences, it seems that the authors just directly copied what I wrote and pasted it into the text, especially in the Conclusions section. This is not advisable, especially because some messages might be misinterpreted when other details are lacking (e.g., population and evolutionary genetics theory, additional info on the heterozygous dogs for the SOD1:c118G>A risk allele and their prevalence of CDM, etc.).
AU: Thank you for your feedback on our manuscript. We apologize for the misunderstanding in our previous response. We now understand your comment and agree with your suggestion. We appreciate the time you took to carefully review our work and provide us with constructive feedback.
The most important concept to stress here is that fact that breeders should avoid the generation of homozygous A/A dogs, given that these are at very high risk of developing CDM. So I would change as follows:
-Line 37-38: I would say "in order to avoid the emergence of dogs homozygous for the SOD1:c118G>A risk allele."
-Line 309-315: I would say "Genetic testing is important to avoid the emergence of dogs homozygous for the SOD1:c118G>A risk allele, as these homozygous dogs will be at very high risk of developing CDM."
AU: We have amended both statements as suggested.
2) It seems that the authors misinterpreted my suggestions on the additional figure representing a correlation plot of the risk allele frequency in all the overlapping (in common) breeds between this study and the study by Zeng et al., 2014.
I would suggest to change Figure 2 for two reasons: a) the current figure is cut, so only one part is visible, and this is not acceptable. b) the current figure is not at all a correlation plot. A correlation plot is a scatter plot, in which the x axis represents in this case the frequency of the risk allele in one study, and the y axis the frequency of the risk allele in the other study. Each overlapping breed then appears as a single point on the graph. Ideally, the plot also shows a regression line that describe the trend of the data, in other words the correlation. I hope it is clear now.
AU: Thank you for the clarification. We have replaced the Figure 2 with a scatter plot as per the instructions provided.
MINOR
1) Line 46-47: I think "CDM was first described in a German Shepherd Dog in 1973" does make little sense. Shouldn't it be "CDM was first described in the German Shepherd Dog in 1973"?
AU: We have changed as suggested.